# Structural basis for broad anti-phage immunity by DISARM

Jack P. K. Bravo[1,8], Cristian Aparicio-Maldonado[2,3,4,8], Franklin L. Nobrega[4], Stan J. J. Brouns [2,3✉] &
David W. Taylor [1,5,6,7✉]

In the evolutionary arms race against phage, bacteria have assembled a diverse arsenal of antiviral immune strategies. While the recently discovered DISARM (Defense Island System Associated with Restriction-Modification) systems can provide protection against a wide range of phage, the molecular mechanisms that underpin broad antiviral targeting but avoiding autoimmunity remain enigmatic. Here, we report cryo-EM structures of the core DISARM complex, DrmAB, both alone and in complex with an unmethylated phage DNA mimetic. These structures reveal that DrmAB core complex is autoinhibited by a trigger loop (TL) within DrmA and binding to DNA substrates containing a 5′ overhang dislodges the TL, initiating a long-range structural rearrangement for DrmAB activation. Together with structure-guided in vivo studies, our work provides insights into the mechanism of phage DNA recognition and specific activation of this widespread antiviral defense system.

[1] Department of Molecular Biosciences, University of Texas at Austin, Austin, TX 78712, USA. [2] Department of Bionanoscience, Delft University of Technology, Delft, The Netherlands. [3] Kavli Institute of Nanoscience, Delft, The Netherlands. [4] School of Biological Sciences, University of Southampton, SO17 1BJ Southampton, UK. [5] Interdisciplinary Life Sciences Graduate Programs, University of Texas at Austin, Austin, TX 78712, USA. [6] Center for Systems and Synthetic Biology, University of Texas at Austin, Austin, TX 78712, USA. [7] LIVESTRONG Cancer Institutes, Dell Medical School, Austin, TX 78712, USA. [8] These authors contributed equally: Jack P. K. Bravo, Cristian Aparicio-Maldonado. ✉email: stanbrouns@gmail.com; dtaylor@utexas.edu

The threat of bacteriophage has driven bacteria to evolve a myriad of antiviral defense systems, capable of targeting phage at various stages of infection and replication[1,2]. The first lines of antiviral defense are typically rapid, broad, innate immune responses that include preventing phage adsorption[3], degrading foreign DNA using restriction-modification (RM) systems[4], or ultimately host cell destruction through abortive infection (Abi) systems[1]. Adaptive immune responses (including CRISPR-Cas systems) provide a highly-specific and long-term antiviral protection[5,6]. A number of recently discovered anti-phage systems have expanded the arsenal of defense tools used by microbes in the arms race against phage[7–9].

RM systems are the most highly abundant bacterial defense system, found in ~75% of all bacterial genomes[10]. They often consist of DNA methyltransferase, restriction endonuclease, and target recognition modules. The vast majority rely on sequence-specific DNA methylation, which is recognized by and rapidly degraded by specific nucleases, allowing for self-versus-non-self discrimination to avoid autoimmunity. However, phage have adapted to evade such systems by either evolving to lack the restriction site sequences required for DNA cleavage, or carrying their own epigenetic modification to prevent being targeted[4,11].

The largely uncharacterized DISARM (Defense Island System Associated with Restriction-Modification) systems are widespread in bacteria[7]. The DISARM operon typically contains a DNA methyltransferase (DrmMI and/or DrmMII that methylate adenine and cytosine, respectively) along with a helicase (DrmA), a DUF (domain of unknow function)1998-containing protein (DrmB), a phospholipase D (PLD) domain nuclease (DrmC), amongst additional auxiliary genes[7] (Fig. 1a). DUF1998 domains are common in various defense systems including Druantia[8] and Dpd[12], suggesting a common role in anti-phage activity. While the presence of methyltransferase and nuclease genes within the DISARM operon hints that this system behaves akin to other RM systems, DrmC appears to be non-essential for DISARM activity, and DISARM can still restrict phage that lack methylation target sequences[13]. Thus, the molecular mechanisms underlying phage targeting remain unclear.

Here, we report cryo-EM structures of DrmA:DrmB (DrmAB) in the presence or absence of target DNA, revealing the arrangement of conserved RecA helicase domains relative to the DUF1998 domain. In vivo studies of structure-guided DrmAB mutants demonstrate that ATP hydrolysis, DNA binding, and DrmAB heterodimer formation are essential for phage targeting by DISARM. We observe that DrmA contains an unstructured trigger loop that partially occludes the DNA binding surface on the complex. This loop limits DNA binding for dsDNA and allows DrmAB to discriminate between targets based on DNA

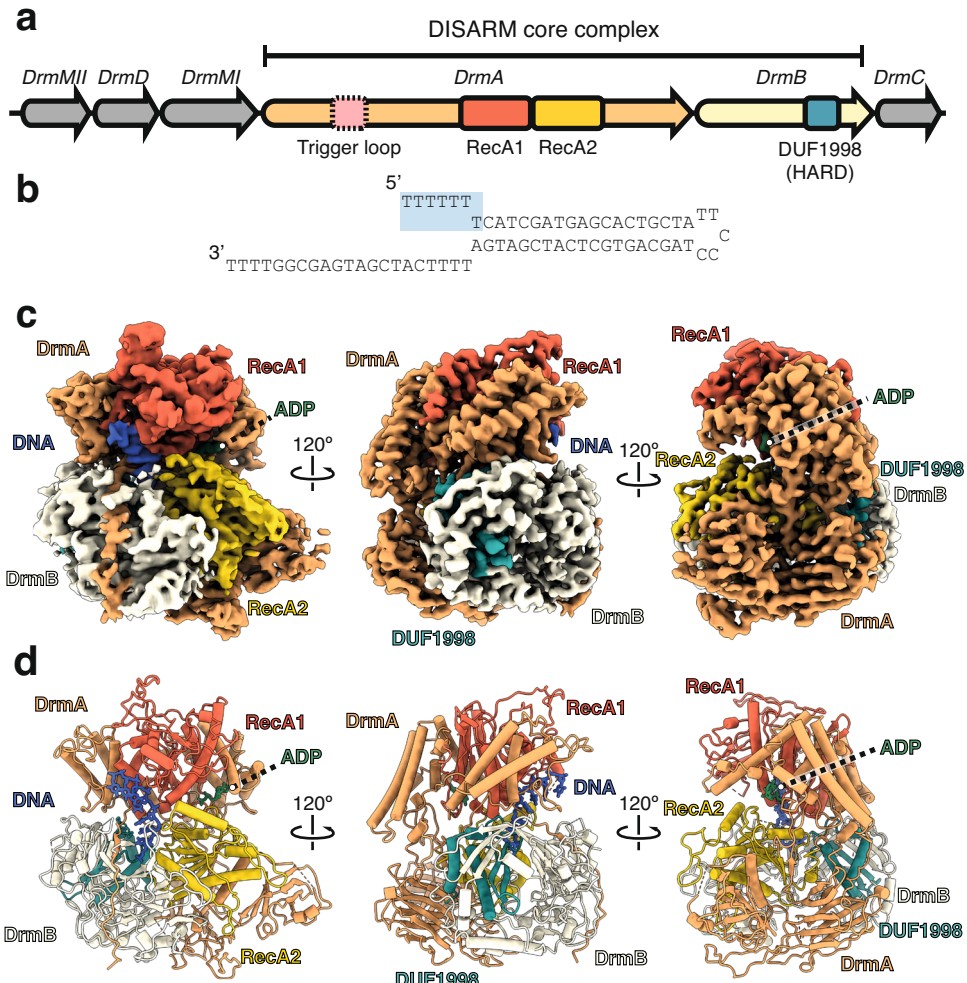

**Fig. 1 Architecture of the DrmAB:ADP:DNA complex. a** *Serratia* sp. DISARM operon. Conserved DISARM core components DrmA and DrmB are colored by structural domains. Presence of DrmA trigger loop (residues 176-232) is mutually exclusive with DNA binding and is denoted by a dashed box. **b** DNA substrate used for structure determination. Seven 5′ bases that are present in our structure are highlighted blue. **c** Cryo-EM reconstruction of DrmAB:ADP:DNA complex colored as in (**a**). ssDNA is shown in blue and ADP in green. **d** Atomic models built into the cryo-EM map (**b**).

structure rather than DNA sequence as a mechanism of self-versus-non-self-discrimination. DNA loading induces long-range conformational changes likely associated with DISARM activation. Our results elucidate how DISARM systems can provide broad anti-phage activity, while avoiding autoimmune activation and reveal key molecular mechanisms that underpin rapid DISARM activation upon phage infection.

## Results

**Architecture of DrmAB nucleoprotein complex.** We purified the *Serratia* sp. SCBI (*Serratia*) DrmAB complex after co-expression of DrmA and DrmB in *E. coli*. SDS-PAGE analysis of the peak fraction from size-exclusion chromatography confirmed the presence of both DrmA and DrmB, in a ~1:1 stoichiometry (Supplementary Fig. 1). Candidate nuclease DrmC did not co-purify with DrmAB when co-expressed, supporting the notion that it does not constitute part of the core DISARM complex[7]. To capture the DrmAB complex bound to a phage single-stranded DNA mimic, we incubated DrmAB with an unmethylated forked DNA substrate in the presence of ADP. DrmAB:ADP:DNA complex formation was confirmed by native electrophoretic mobility shift assays (Supplementary Fig. 1).

We used cryo-EM to investigate the architecture of the DrmAB:ADP:DNA complex. During data collection, we observed that the complex adopts a preferred orientation in vitreous ice, which was ameliorated through collecting additional data at a −30° tilt. Multiple rounds of classification resulted in a 3D reconstruction of the complex at a nominal resolution of 2.8 Å and focused 3D classification improved densities corresponding to DNA and the N-terminus of DrmB. Notably, we were able to separate DNA-bound from DNA-free particles through focused 3D classification, ultimately yielding 3D reconstructions of DrmAB:ADP:DNA and DrmAB:ADP at nominal resolutions of 3.3 and 3.4 Å, respectively (Supplementary Fig. 2). Between these two structures, we determined atomic models that account for >98% of the total residues of the ~220 kDa protein components, with an additional seven nucleotides of DNA in the DrmAB:ADP:DNA complex (Supplementary Fig. 3).

The DrmAB complex exhibits a bi-lobed architecture, resembling a partially open clam shell (Fig. 1b, c). The top lobe contains the RecA1 domain of DrmA, while the bottom lobe contains the DrmA RecA2 domain and the entirety of DrmB.

As expected for a superfamily 2 (SF2) helicase, DrmA contains tandem RecA modules (residues 600–1150) that form an active site for ATP-binding and hydrolysis and DNA loading. The N-terminal half of DrmA acts as a structural chassis, holding these two helicase motor domains in place, and forming an interface with DrmB. While RecA1 and RecA2 have high structural similarity to many other SF2 helicase domains (Z-score >14 for more than 10 other helicase structures[14]), no other regions of DrmA were found to have significant structural homology to known protein domains. The solvent-exposed interface between the tandem RecA domains forms the nucleotide-binding site, while the bound DNA straddles these domains within the heart of the complex (Fig. 1b, c). Only four of the seven deoxynucleotides observed within our structure are solvent-exposed, with the remaining three bases deeply buried within the DrmA(RecA1,RecA2):DrmB interface.

DrmB contains a C-terminal DUF1998 module, a domain often enriched within bacterial genomic defense islands[7,15]. DUF1998 domain sits towards the back of the complex, predominantly buried within DrmB, positioned towards the interface between the two RecA domains of DrmA. Like other proteins containing DUF1998 domains, DrmB contains the predicted zinc-coordinating four cysteine motif (C559,565,581,584, Supplementary Fig. 4)[16,17].

Surprisingly, we also observed an additional four putative coordinated metal ions at highly conserved sites within DrmB (Supplementary Fig. 4). While it is not possible to unambiguously determine the identity of these ions, based on the coordinating residues $Zn^{2+}$ is a strong candidate[18].

This raises the possibility that DrmAB may utilize these three highly conserved metal ion coordination sites to sense changes to cellular redox potential, triggered by environmental stresses such as phage infection, as has recently been demonstrated for the type III-A CRISPR system within *Serratia*[19–22]. Alternatively, these clusters may play a structural role, enabling proper folding of DrmB, as is the case for the DUF1998-containing StfH protein[17].

Multiple interaction surfaces contribute to the stabilization of the DrmAB complex, with a total buried surface area of ~4000 Å². Most notably, the C-terminus of DrmA (residues 1296–1319) wraps around the entirety of the N-terminus of DrmB (Fig. 1b, c), with a total buried surface area of ~1850 Å². Additionally, the DrmB DUF1998 domain mediates ~1230 Å² of surface contacts with DrmA (Fig. 1d). DUF1998 is positioned towards the interface between the two RecA domains of DrmA but makes limited contacts with both (~130 and ~110 Å² buried surface area with RecA1 and RecA2, respectively). RecA2 makes additional surface contacts with the N-terminus of DrmB (482 Å²). Based on this network of inter-subunit interactions, we hypothesize that DUF1998-containing DrmB may act as a modulator of DrmA helicase activity. This is supported by the observation that DUF1998 domains are frequently associated with helicase domain-containing proteins, either as adjacent genes or as C-terminal fusions[15].

**DNA binding requirements of DrmAB.** The DNA-bound DrmAB complex includes a 7-mer ssDNA segment (Fig. 2a). While this is only a fragment of the DNA used for complex assembly, comprising a 19-bp duplex with a 5-nucleotide hairpin on one end and a 7-nucleotide 5′ tail and a 21-nucleotide 3′ tail on the other end, we attributed this to the flexibility of DNA that does not make direct contact with the complex. We tested the possibility of DrmAB requiring a 7-nt overhang to bind DNA by measuring the binding of DrmAB to DNA substrates with various 5′ overhang lengths. We observed binding of DrmAB to a 7-nt 5′ ssDNA overhang, and minimal binding to substrates with fewer than 7-nt overhangs (Supplementary Fig. 1e).

DrmA has multiple interactions with the backbone of the DNA, mostly electrostatic contacts between positively charged side chains and DNA phosphate groups (Fig. 2b, c). Such non-specific electrostatic DrmAB:DNA contacts provide the molecular basis for the previously observed broad anti-phage targeting by DISARM[7]. Since conventional RM systems require recognition of specific DNA sequence motifs for nuclease activity, they can be easily evaded through phage evolving escape mutations[11,23]. However, by lacking sequence preferences for DNA binding, DISARM can provide effective anti-phage defense in the absence of a particular restriction site.

To address the functional relevance of these interactions, we mutated various DNA-interacting residues within DrmA (K803, R1294, R659, R810) to alanine. All four mutants showed reduced anti-phage protection in vivo (Fig. 2d). Together, these data underscore the functional role of DrmA as the DNA targeting arm of the complex and provide the mechanism of targeting a wide range of phage by DISARM. Overall, the non-specific interactions between DrmA and DNA provide a structural basis for broad phage targeting by DISARM.

**ATPase activity is critical for DISARM function.** We observed strong density corresponding to ADP within our DNA-bound

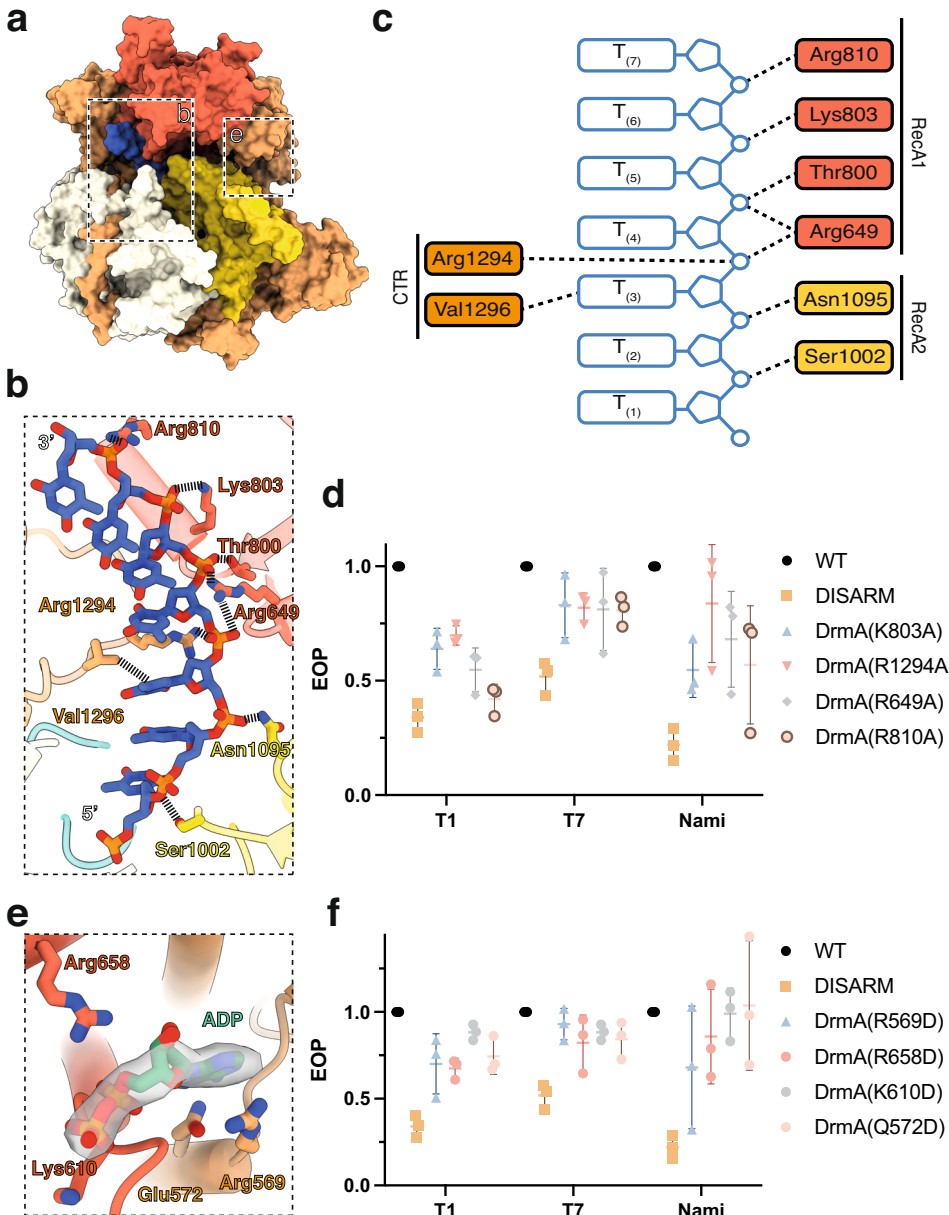

**Fig. 2 Both ssDNA and nucleotide cofactor binding are essential for DISARM function. a** Overall structure of DrmAB:DNA:ADP. Boxes indicate regions featured in panels (**b**, **e**). **b** Close-up view of DrmA:DNA interactions. No interactions between DrmB and DNA are observed. **c** Schematic of DNA-protein contacts colored by protein domain. **d** Effect of mutagenesis of DrmA DNA-interacting residues on DISARM anti-phage activity. Points correspond to three biological replicates, with mean and standard deviation shown. Source Data are provided as a Source Data file. **e** Close-up view of DrmA:ADP interactions. **f** Effect of mutagenesis of DrmA ADP-interacting residues on DISARM anti-phage activity. Points correspond to three biological replicates, with mean and standard deviation shown. Source Data are provided as a Source Data file.

(Fig. 2e) and DNA-free maps, both of which were at the interface between the RecA1 and RecA2 domains, the canonical ATP-binding site of SF2 helicase domains. While the RecA domains of DrmA suggest that it is a helicase, the impact of its helicase activity on anti-phage targeting is poorly understood. We mutated four ADP-interacting residues to test their impact on DISARM activity. All four mutants showed reduced DISARM activity against a broad range of phage in vivo. This confirmed that in addition to DNA binding, ATP hydrolysis by DrmA is essential for DISARM activity. This is similar to type I RM systems, which encode a DEAD-box helicase domain protein that drives DNA translocation upon recognition of an unmethylated restriction site[10,24]. Akin to DrmAB, ATP-dependent DNA translocation is essential for type I RM system activity[25]. However, it is unlikely

that these two anti-phage systems share a common DNA degradation mechanism, since type I RM systems translocate and cleave dsDNA, while DISARM appears to bind exclusively ssDNA (Fig. 3).

**DrmA contains an unstructured trigger loop that partially occludes the DNA-binding site**. Two dominant, distinct populations (~120,000 particles each) emerged during focused 3D classification; one of which had strong DNA density (DrmAB:ADP:DNA) and one lacking observable DNA density (DrmAB:ADP). Comparison of our two maps revealed the presence of a 55-residue loop in DrmA that is otherwise absent in the DNA-bound structure (Fig. 3a, b). Due to the flexibility of this loop, we

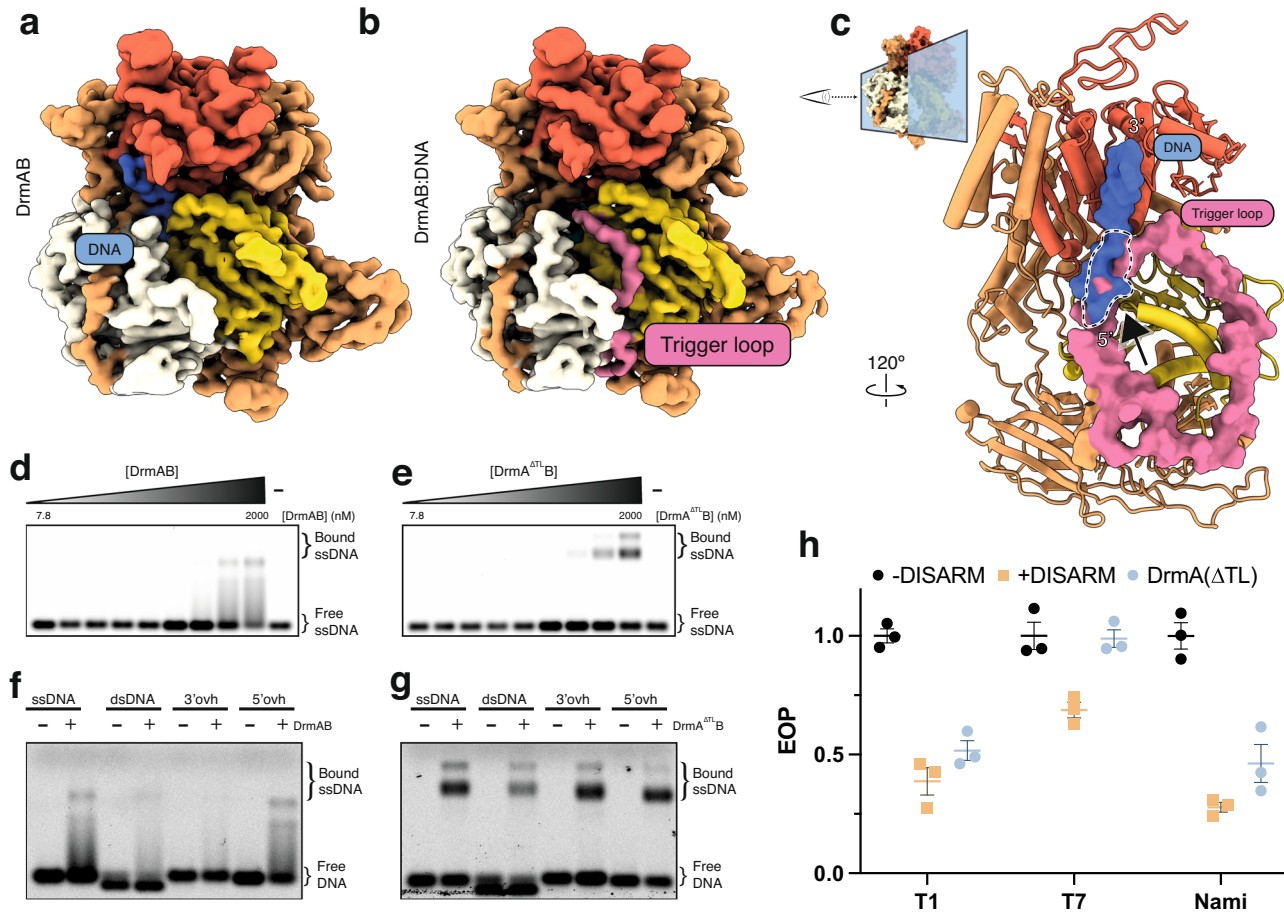

**Fig. 3 DrmA trigger loop (TL) partially occludes DNA-binding site. a, b** Cryo-EM reconstruction of apo DrmAB (**a**) and DNA-bound DrmAB (**b**). Density corresponding to DrmA trigger loop is shown in pink. DrmAB are colored by structural domains as shown in Fig. 1a. **c** Overlay of DrmAB-bound DNA (blue) and TL (pink) showing the steric clash (dashed black & white outline). TL partially occludes DNA-binding site. DrmB and parts of DrmA have been omitted for clarity. Graphic at the top right shows how the view in (**c**) is related to the structures in panels (**a, b**). **d, e** EMSA analysis of DrmAB and DrmA(Δloop)B binding DNA. The smeared bands that occur at high concentrations of DrmAB correspond to bound DNA dissociating from the complex. This did not occur for DrmA(Δloop)B, and an additional super-shift was present, corresponding to multiple copies of DrmA(Δloop)B binding to the same 75-nt DNA concurrently. The free DNA marker is 75-nt in length. Representative of three independent experiments. Source Data are provided as a Source Data file. **f, g** DNA substrate preferences of DrmAB and DrmA(Δloop)B. The 75-nt Cy5-labeled DNA (ssDNA) was annealed to complementary oligos corresponding to the full sequence (dsDNA), or 20 bases at the 5′ or 3′ ends (3′ovh and 5′ovh, respectively). DrmAB showed a preference for ssDNA and 5′ovh DNA, whereas DrmA(ΔTL)B bound to all DNA substrates tested. The free DNA marker is 75-nt in length. Representative of three independent experiments. Source Data are provided as a Source Data file. **h** Effect of truncation of TL on DISARM anti-phage activity. Points correspond to three biological replicates, with mean and standard deviation shown. Source Data are provided as a Source Data file.

were unable to confidently assign sequence position. We modeled this region as poly-alanine. To further validate this model, we determined a structure of DrmAB in the absence of DNA and ADP at a resolution of 3.8 Å (Supplementary Fig. 5). Rigid-body docking of our higher-resolution DrmAB:ADP model revealed density consistent with the presence of this loop (Supplementary Fig. 5). Thus, the presence of this loop is mutually exclusive with DNA binding.

After superposition of our two models, we were surprised to observe a severe steric clash between DNA and this unstructured loop (Fig. 3c) that occurs at the interface between the 5′ DNA end and RecA2. We hypothesized that this region of DrmA acts as a trigger loop (TL) (residues 176–232) that activates DrmAB after being dislodged by target DNA. Given that the presence of TL and DNA are mutually exclusive, we hypothesized that in the absence of single-stranded DNA, this loop may partially occlude the DNA binding site, preventing DrmAB from loading onto non-phage (i.e., host) DNA. To study its relevance in the DISARM mechanism, we created a mutant lacking this loop

(DrmA(Δloop)), removing residues 181–233. Since DrmA(ΔTL) maintained its ability to co-purify with DrmB, and the DrmA(Δloop)B complex eluted at the expected volume according to size-exclusion chromatography (Supplementary Fig. 1a), heterodimer assembly does not depend on this region of DrmA. We performed native electrophoretic mobility shift assays to compare binding of DrmAB and DrmA(ΔTL)B to fluorescently-labeled 75-nt single-stranded DNA (Fig. 3d, e). While DrmAB bound DNA at concentrations of 1 μM or higher, the presence of smeared bands likely indicated dissociation of DNA from the complex. In contrast, DNA dissociation from DrmA(ΔTL)B bound DNA was not observed. Furthermore, a second super-shift occurred at protein concentrations of 1 μM or higher, likely corresponding to multiple copies of the complex binding to the DNA concurrently, as observed for other nucleoprotein complexes[26].

Since TL limits DNA binding, we performed additional binding assays to determine how this loop affects DNA substrate preferences of DrmAB (Fig. 3f, g). DrmAB bound to ssDNA and

DNA with a 5′-overhang (ovh) but did not bind to dsDNA or DNA with a 3′ovh. DrmA(ΔTL)B displayed no DNA substrate preference, binding to ssDNA, dsDNA, and DNA with both a 5′ovh and 3′ovh. Thus, the TL limits structure-specific DNA binding by partially occluding the DNA-binding surface of DrmAB.

While the methylation modules of the DISARM operon recognize specific sequences, DISARM targets a diverse range of phage DNA sequences[7]. We thus propose that TL functions as a specificity filter, preventing interactions with dsDNA and DNA containing 3′ovh. Since DNA would occupy a larger surface on DrmAB than TL, DNA with a 5′ovh would be an ideal substrate for efficiently competing with TL for loading onto the complex. This may function as a mechanism for conferring substrate specificity to DISARM and prevent interactions with bacterial chromosome and other self-DNA, which would likely result in deleterious autoimmune effects. By relying on DNA structural context rather than simply DNA sequence, DISARM appears to utilize an alternative mechanism for avoiding self-targeting to many other nucleic acid-based anti-phage systems[5,24].

**Methylation sensing and DNA-mediated DrmAB activation.** We sought to visualize additional conformational changes within DrmAB associated with DNA loading and complex activation. Within the core DNA-binding site of DrmAB, continuous cryo-EM density is observed between the DNA nucleobase at position 3 ($T_{(3)}$, Fig. 2c) and the side-chain of DrmA(V1296), likely corresponding to a van der Waals interaction (Fig. 4a). Since this residue is highly conserved (found in >98% of top 250 DrmA homologs) and is the sole interaction between DrmAB and a DNA base, we hypothesized that DrmA(V1296) may act as a sensor for DNA methylation status. *Serratia* DISARM operon contains both a 5-cytosine DNA methyltransferase and an N6-adenine DNA methyltransferase gene which target ACAC(m**A**)G and YMT(m**C**)GAKR motifs, respectively (Fig. 1a), which may constitutively methylate the host genome to allow distinguishing self (i.e., methylated) from non-self (i.e., unmethylated) phage DNA, akin to RM systems[7,11].

Further analysis of our structure revealed that DrmA(V1296) is in close contact with the faces of the DNA nucleobases, tightly wedged between DNA positions 3 and 4. Modeling this position as a bulky tryptophan (W) side-chain showed severe steric clashes, and this mutant rendered DISARM unable to protect against phage in our in vivo assay (Fig. 4b). We then tested the in vivo anti-phage activity of DISARM with a glycine at this position (DrmA(V1296G)), which would no longer contact DNA nucleobases. This also resulted in severely reduced DISARM activity, supporting the notion that V1296 plays a critical role in DNA recognition (Fig. 4b). We propose that since this interaction is critical to DISARM function, differences in DNA methylation status may perturb this interaction, providing a structural mechanism for enabling DISARM to differentiate between self- and non-self DNA and preventing autoimmune targeting.

To further test this hypothesis, we performed ATPase assay of DrmAB in the presence and absence of methylated and unmethylated DNAs. In the absence of DNA, DrmAB did not exhibit ATPase activity (Fig. 4b). This is expected for a SF2 helicase, where DNA translocation is typically coupled to ATP hydrolysis[27]. Robust ATPase activity is strongly stimulated by unmethylated DNA, but the ATPase rate is reduced in the presence of a methylated DNA substrate (containing three separate 5-methyl-cytosine bases, i.e., the modification provided by DrmMII[7]. This supports the model that this multi-protein complex detects phage DNA as it is injected into bacteria or is being replicated. Additionally, recent studies demonstrate that

DISARM can protect against plasmid conjugation, and that DISARM targeting is enhanced by an order of magnitude in the absence of cognate methylation target sequences[28]. This data indicates that DISARM is repressed by methylation rather than being activated by the presence of a specific signal sequence.

We then compared our DNA-bound and DNA-free DrmAB complexes by superposing our models and generating motion vectors (Supplementary Fig. 6). Upon loading of unmethylated DNA, we observed minor (mostly < 5 Å) conformational changes of the RecA1 and RecA2 domains (Fig. 4a). Surprisingly, we observed large conformational rearrangements in the N-terminal half (NTH) of DrmB, despite a lack of any contacts with DNA. DrmB NTH shifts ~10 Å towards the RecA2 domain of DrmA, tightly clamping the complex around DNA.

This long-range allosteric communication appears to be mediated by the C-terminal region of DrmA, which tightly wraps around the N-terminus of DrmB acting as a Pivot Arm (PA). While DrmA(V1296) undergoes a minor shift upon recognition of unmethylated DNA, this is propagated into a much larger conformation change in the end of the PA and DrmB NTH. Thus, conformational change upon loading unmethylated DNA activates DrmAB. Since the DUF1998 domain of DrmB acts a hinge for the conformational changes induced by the PA, we propose that DUF1998 should be renamed as a Helicase Allosteric Relay (HAR) domain.

DrmA PA is tightly interwoven with DrmB NTH, forming a myriad of contacts. These include a network of electrostatic interactions, hydrophobic contacts and π-π stacking (Fig. 4d). We also observed that a five-residue segment of DrmA PA contributes to a 5-stranded β-sheet in *trans* with DrmB, forming an oligonucleotide/oligosaccharide-binding (OB) fold (Fig. 4e). The β-strand from DrmA is sandwiched between strands from DrmB, further locking the PA of DrmA in place. These intimate contacts allow DrmA to allosterically communicate with DrmB, setting in motion major conformational changes upon loading of unmethylated DNA.

To test the importance of this conformational change for DISARM activity, we truncated DrmA so that it lacked the PA and tested the ability of DISARM to prevent phage replication in vivo. We found that DrmA-ΔPA had severely reduced DISARM activity against a broad range of phage, supporting the notion that the CTR-driven conformational changes of DrmAB result in DISARM activation (Fig. 4f). Based on our structural and functional data, we propose that upon displacement of the TL by ssDNA, DrmA(V1296) senses the DNA methylation status, allosterically activating DrmAB and triggering the DISARM anti-phage response.

In summary, through coupling ATPase activity to DNA nucleobase methylation status, DrmAB may be able to achieve specific targeting of unmodified non-self (i.e., phage) DNA without relying on a given DNA sequence.

## Discussion

We propose a model whereby DrmA and DrmB are expressed by the host cell in an autoinhibited form in the absence of phage infection (Fig. 5). Through the constitutive expression of the core DrmAB complex, a DISARM response can be rapidly activated upon recognition of phage stimulus without necessitating transcription of either component, in accordance with previous observations[7]. In the absence of DNA, the DNA-binding channel of DrmAB is occluded by an unstructured ~50 residue loop within TL. While truncation of this loop does not affect heterodimer formation, we observed that DrmA(Δloop)B no longer exhibited discrimination for 5′ovh-containing DNA substrates and could bind both 3′ovh and dsDNA. We conclude that this loop

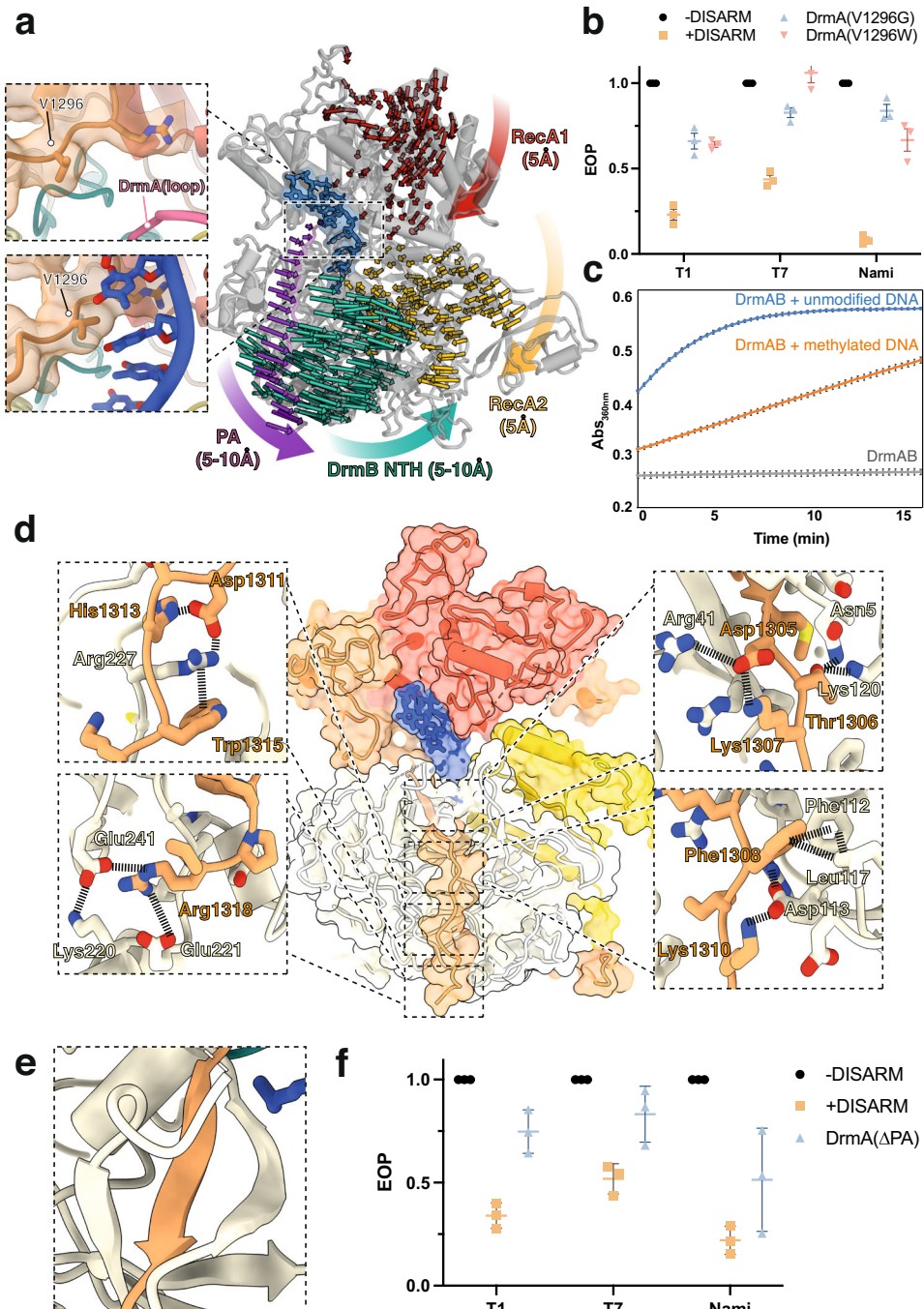

**Fig. 4 Binding of unmethylated ssDNA activates DrmAB through large conformational rearrangements. a** DrmA C-terminal pivot arm (PA) interface with substrate DNA. Left: close-up views of DrmAB map and model lacking DNA (top) and DrmAB:DNA map and model (bottom). In the DNA-bound map, a direct interaction between a DNA base and Val1296 is present (black arrow). In the DrmAB map lacking DNA, the DrmA autoinhibition loop is present (pink arrow). Right: Motion vectors showing conformational changes of DrmAB upon DNA binding. DrmB N-terminal half (DrmB NTH, cyan) does not make direct contacts with DNA but undergoes a large conformational change upon DNA binding, mediated by the PA of DrmA (magenta). Distances of conformational changes are shown parenthetically. Motion vectors for shifts smaller than 3 Å are omitted. **b** EOP assay to determine the effects of V1296G or V1296W mutations to DISARM activity. Points correspond to three biological replicates, with mean and standard deviation shown. Source Data are provided as a Source Data file. **c** ATPase assay measuring the rate of ATP hydrolysis by DrmAB, as monitored by increase in absorbance at 360 nm. In the absence of DNA, no ATPase activity is monitored. In the presence of DNA stem loop lacking modification (blue), ATPase activity is observed. This is severely reduced for DNA containing three separate 5-methyl cytosine bases (orange). DNA sequences are listed in Supplementary Table 2. Experiments were performed in triplicate, and the mean and standard deviation of the mean are depicted. Source Data are provided as a Source Data file. **d** Close-up views of DrmA(CTR):DrmB interactions. **e** DrmA C-terminus forms a β-sheet in *trans* with DrmB. **f** Effect of truncation of DrmA PA on DISARM anti-phage activity. Points correspond to three biological replicates, with mean and standard deviation shown. Source Data are provided as a Source Data file.

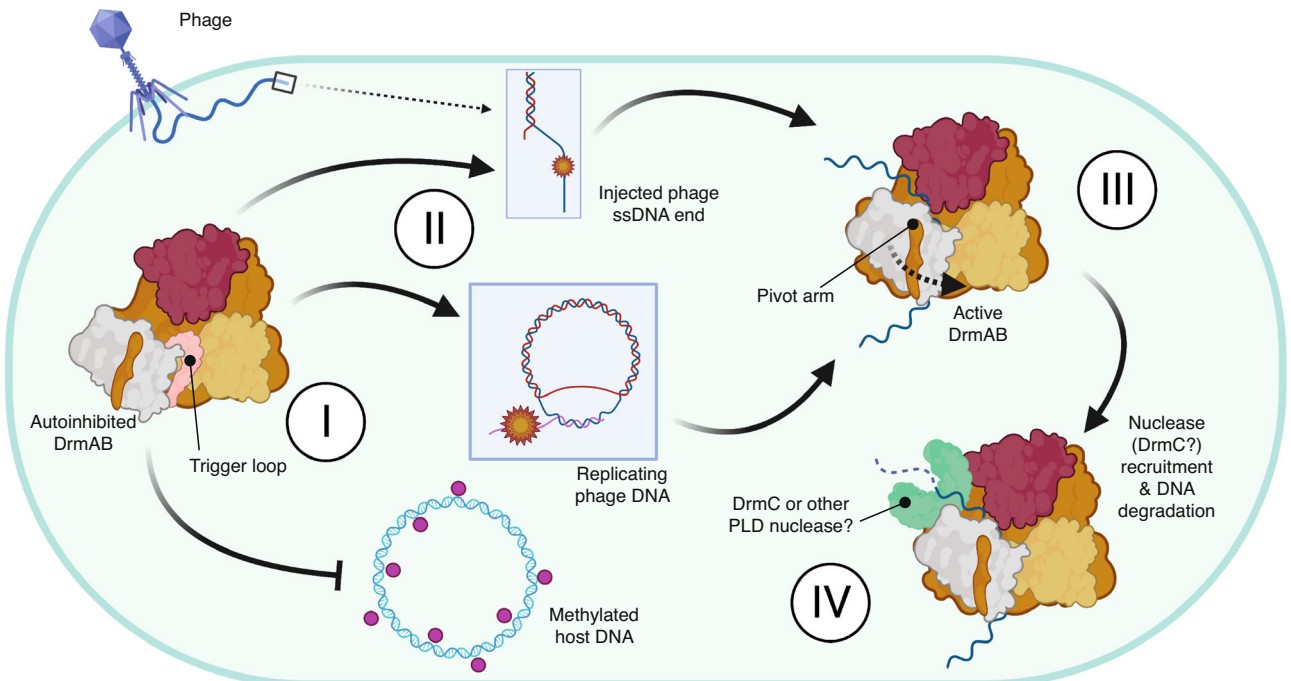

**Fig. 5 Model for anti-phage surveillance by DISARM. I** In the absence of phage infection, DrmAB complex is autoinhibited via TL. This allows the constitutive expression of the complex to enable rapid immune activation upon stimulus. **II** DISARM is recruited to single-stranded 5′ ovh DNA, which typically occur during initial phage DNA injection prior to genome circularization, or during rolling circle DNA replication. **III** Loading of ssDNA into DrmAB dislodges TL, resulting in a conformational change and DrmAB activation. **IV** Once DrmAB is active, DrmC or other nucleases may be recruited to degrade foreign DNA. DISARM may act to defend against phage at step III by loading onto phage DNA ends and physically blocking replication, and at step IV through recruiting a nuclease and degrading phage DNA. This cartoon was created with Biorender.com.

may act as a selectivity filter or TL, ensuring the loading of specific substrates onto DrmAB. Since TL only partially occludes the DNA binding site, loading of 5′ovh DNA can effectively compete against and evict TL, which is likely intrinsically disordered. A similar mechanism of autoinhibition was recently reported for human Separase[29], and examples of substrate-alleviated autoinhibition are numerous amongst other SF2 helicase complexes[30–33]. Interestingly, a recent study of the human DEAD-box helicase DHX37 revealed a highly similar autoinhibition mechanism, whereby an unstructured protein loop occupied the substrate-binding channel[34]. It may emerge that similar mechanisms of autoinhibition are widespread amongst a diverse group of SF2 helicase proteins from all domains of life.

Based on our data, we propose that the activating substrate for DISARM is unmodified 5′ovh DNA. Unlike many RM systems, DISARM does appear to not rely on specific sequences, enabling broad anti-phage targeting that cannot be circumvented through escape mutation. This is further supported by in vivo functional data that show that DISARM activity is enhanced by a lack of cognate methylation target sequences[28]. Future studies are required to investigate the interplay between DrmAB and other less-conserved DISARM components (e.g., DrmD and DrmE) within DISARM, and how these subunits affect the specificity of DISARM activation.

The preference for 5′ovh ssDNA is particularly important since many dsDNA phage inject their genome in a linearized form with two sticky end 5′ovh, which subsequently become ligated during DNA circularization[35,36], thus providing a stimulus for DISARM activation. This is supported by the previous observation that DISARM has no effect on phage adsorption but blocks phage DNA circularization[7]. If a dsDNA phage does not have such 5′ ovh, then it would still likely be targeted by DISARM since it would still replicate via rolling circle DNA replication, where the

leading strand is replicated in a 5′ to 3′ direction and the lagging strand is synthesized as Okazaki fragments[37]. This model explains the previous observation that DISARM is activated by the onset of phage replication, likely resulting probably in the degradation of viral DNA[7]. This data also explains how DrmAB target specificity is conferred by DNA structure rather than a specific sequence, providing an elegant mechanism for balancing broad DNA targeting with minimal autoimmune consequences. DISARM therefore represents a novel paradigm in bacterial antiviral defense mechanisms that target nucleic acids. Rather than achieving specific and efficient degradation of phage DNA through recognition of specific sequences, DISARM recognizes the structural context of DNA. The core DISARM complex DrmAB is able to confer specificity of DNA loading by targeting substrates with a 5′ovh, thereby avoiding degradation of the host chromosomal DNA. This specificity is further conveyed by a sensitivity to DNA methylation status, whereby methylated DNA limits the ATPase activity of DrmAB which is essential for anti-phage defense. This amounts to a two-pronged mechanism of selectivity that does not rely on a given DNA sequence, and thereby circumvents evasion via mutational escape.

While this model is consistent with our data, the class 2 DISARM system has been demonstrated to confer protection against modified phage[7]. While DISARM preferentially targets unmodified DNA, methylated DNA can still support ATP hydrolysis and thus defense activation. It may be the case that DISARM activation is significantly reduced by phage DNA methylation, but the abundance of phage 5′ovh during successive cycles of replication may provide sufficient stimulus to activate DISARM. Additionally, other DISARM subunits may confer additional mechanisms to detect invading phage.

We hypothesize that once activated, DrmAB binds to, translocates, and unwinds phage DNA, blocking replication and

transcription and disrupting the phage infection cycle. Additionally, DrmAB may recruit a nuclease to trigger DNA degradation. Since the PLD-containing candidate nuclease DrmC is dispensable for DISARM activity and is often missing from annotated DISARM operons, it has remained a mystery how this is achieved[7]. However, since PLD-containing nucleases are highly abundant in bacterial genomes, we propose that DrmAB may recruit an alternative PLD-nuclease protein. While previous in vivo assays were performed in *Bacillus subtilis* (which encodes a 42.2% identity DrmC-like homolog), our in vivo assays were performed in *Escherichia coli* (which encodes a 93.4% identity DrmC-like homolog). The ubiquity and versatility of such DrmC-like non-specific PLD nucleases[38–41] may make DrmC redundant, allowing DISARM to 'mix-and-match' components from other innate immune systems encoded by the host bacteria in order to achieve rapid and broad anti-phage protection. While CRISPR-Cas systems rely on RecBCD and other host DNA double-strand break repair complexes for adaptation (i.e., acquiring spacers)[42–44], it is unlikely that DISARM utilizes this mechanism, since it does not require spacer acquisition of phage target sequences.

Phage therapy represents a promising avenue to treat bacterial infection in an age where antibiotic resistance is widespread[45]. Since DISARM is an incredibly widespread mechanism of antiviral defense in bacteria, once we understand the fundamental mechanisms underlying DISARM activation it may be possible to develop small molecule treatments to inhibit DISARM, which could be delivered in conjunction with phage therapy to increase the effectiveness of such antibacterial treatment.

## Methods

**Cloning and protein expression**. Genes *drmA* and *drmB* were amplified from genomic DNA of Serratia sp. SCBI using primers BN1750, BN1751, BN1060 and BN1061 (Supplementary Table 3) with Q5 DNA Polymerase (NEB: M0491S) as indicated by the manufacturer. Amplified gene *drmB* was cloned in plasmid 13S-S (Addgene: 48329), resulting in a N-terminal 6X His tagged to the protein product. His-tag was separated from the gene with a tobacco etch virus (TEV) protease recognition sequence. Amplified gene *drmA* was cloned in plasmid pACYCDuet-1 (Sigma-Aldrich: 71147), resulting in a non-tagged protein product. Cloning products were transformed in Dh5α using the heat-shock protocol[46] and confirmed by sanger sequencing (Macrogen).

Both plasmids were transformed into *E. coli* BL21-AI™ (Thermo Fisher) to express the DrmAB complex. Briefly, cultures were grown in LB medium at 37 °C with shaking until exponential growth phase (OD$_{600}$ of 0.5). After cooling down in ice for 20 min, they were induced by addition of 0.2 % (w/v) L-arabinose and 1 mM IPTG. After overnight incubation at 20 °C with shaking, cells were pelleted by centrifugation, resuspended in Buffer A (25 mM HEPES, pH 8, 600 mM NaCl, 5% glycerol, 25 mM imidazole, cOmplete™ EDTA-free Protease Inhibitor), and sonicated (10 cycles of 30 s separated by 30 s breaks, 40% amplitude). Cell lysate was pelleted down (20,000 × *g*, 60 min, 4 °C) to remove cell debris, and supernatant was filtered with a 0.45 μm PES filter and placed on ice until proceeding with the purification.

All the mutants used here were generated by round-the-horn side-directed mutagenesis and purified as the wild-type (WT) proteins.

**Protein purification**. Purification was performed at 4 °C using an ÄKTA™ pure (GE Healthcare) to control flow and column pressure. Lysates were loaded onto a 5 mL Ni-NTA Superflow Cartridges (Qiagen) equilibrated with Buffer B (25 mM HEPES, pH 8, 600 mM NaCl, 5% glycerol, 25 mM imidazole, 1 mM DTT). Unbound lysate components were washed with 10 column volumes (CV) of Buffer B. Bound proteins were eluted with 5 CV of Buffer C (25 mM HEPES, pH 8, 600 mM NaCl, 5% glycerol, 150 mM imidazole, 1 mM DTT) and collected in fractions. After checking by SDS-PAGE electrophoresis using a 4–20% Mini-PROTEAN® TGX precast gel (Bio-Rad), fractions containing DrmAB were pooled together and buffer exchanged to Buffer D (25 mM HEPES, pH 8, 150 mM NaCl, 5% glycerol, 1 mM DTT). To separate the DrmAB complex from proteins that also bound the Ni-NTA column, sample was loaded onto a HiLoad 16/600 Superdex 200 (GE Healthcare). Column was washed with 2 CV of Buffer D and fractions collected. After SDS-PAGE electrophoresis check, fractions containing the complex DrmAB were pooled together and concentrated using 30 kDa NMWL Ultra-15 Amicon® (Merck). Protein concentration was estimated by the Bradford Assay (Thermo Scientific: 23246) as indicated by the manufacturer.

**Bacterial strains used in phage assays**. DISARM system genes from *Serratia sp*. SCBI were cloned in plasmids (Supplementry Table 4), for both WT and mutant. Plasmids were transformed in *E. coli* BL21-AI. For assays, strains were cultured in LB media at 37 °C, and induced with 0.1% (w/v) L-arabinose and 0.5 mM IPTG at early exponential growth phase (OD$_{600}$ of 0.2–0.3). Cultures were incubated at 37 °C for 90 min and then used for phage assays. In parallel, a control strain (BL21-AI WT) was grown for all assays. When required, LB media was supplemented with antibiotics at the following final concentrations: 100 μg/mL ampicillin, 50 μg/mL kanamycin, 50 μg/mL streptomycin, 50 μg/mL spectinomycin, 10 μg/mL chloramphenicol.

**Phage strains**. *E. coli* phages T1, T7, and Nami were used in this study. For their production, phages T1 and T7 were propagated in *E. coli* BL21-AI, as described previously[47], and bacteriophage Nami in its host *E. coli* isolate R10256 following same procedure. Briefly, bacterial cultures at exponential growth phase (aprox. 0.4 OD$_{600}$ in a 10 mm cuvette readers) were infected with a phage lysate and incubated overnight. Then, cultures were spun down and the supernatant filtered through 0.2 μm PES membranes. When required, phages were concentrated by addition of PEG-8000 and NaCl at final concentrations of 100 mg/mL and 1 M, respectively. Following, it was incubated overnight at 4 °C, centrifuged at 11,000 × *g* at 4 °C for 60 min, and the phage-containing pellet resuspended in the desired final volume of Saline Magnesium (SM) buffer. Phage stocks were stored at 4 °C before their use and titer determined as indicated below.

**Phage titering**. Bacterial cultures in exponential growth phase were used to titer phages stocks. For this, 100 μL of culture were mixed with 5 mL of 0.6% LBA at 45 °C and poured to a LBA plate to form a bacterial layer. Ten-fold dilution of phages in SM buffer or LB media were plated on the top of the bacteria in 10 μL drops and let dry for 20 min. Plates were incubated overnight up-side down at 37 °C. To determine the phage titer, the number of center of infections (plaques) were counted. *E. coli* BL21-AI was used for titering phages T1, T7, and Nami.

**Efficiency of plating determination**. To determine the efficiency of plating (EOP), phages were plated on the induced strains containing the DISARM system and compared to the plating on control strain BL21-AI. For this, 200 μL of the bacterial cultures were mixed a specific, countable number of infectious particles (50–150 PFU/plate) and 4.5 mL of 0.6% LBA prewarmed at 45 °C. The mixture was then poured on top of a LBA plate to form a bacterial layer containing the infectious particles. After overnight incubation at 37 °C, the EOP was calculated by dividing the number of plaques counted on each plate by the number of plaques formed in the control strain.

**CryoEM sample preparation, data collection and processing**. To capture DrmAB in the act of unwinding a DNA substrate, a forked DNA construct consisting of a 19-bp stem-loop structure with a 6 base 5′ overhang and a 20 base 3′ overhang was designed. This substrate was chosen because similar substrates have been used to capture unwinding intermediates of bacterial SF2 helicase complexes[48,49]. DrmAB was buffer exchanged into cryoEM sample buffer (150 mM NaCl, 25 mM HEPES pH 7.4) through repeated application of buffer to sample within a 0.5 ml spin concentrator, with a 30 kDa size cutoff. 10 μM DrmAB was incubated with 10-fold excess DNA stem loop (100 μM) and 100-fold excess ADP (1 mM). Complex formation was monitored using electrophoretic mobility shift assay (EMSA), confirming DNA binding by DrmAB (Supplementary Fig. 1). After incubation at room temperature (~25 °C) for 30 min, DrmAB:ADP:DNA was applied to glow discharged holey carbon grids (C-flat 2/2, Protochips Inc.), blotted for 2 s with a blot force of 4 and rapidly plunged into liquid ethane using an FEI Vitrobot MarkIV.

Data was collected on an FEI Titan Krios cryo-electron microscope equipped with a K3 Summit direct electron detector (Gatan, Pleasanton, CA) operating in super-resolution counting mode. Images were recorded with SerialEM v3.8[50] with a pixel size of 1.1 Å over a defocus range of −1.5 to −2.5 μm. During early stages of data collection, a preferred orientation was observed. To ameliorate this, a dataset of 6828 micrographs was collected at 30° tilt, in addition to the original 2548 micrographs collected without tilt. Movies were recorded at 13.3 electrons/pixel/second for 6 s (80 frames) to give a total dose of 80 electrons/pixel. CTF correction, motion correction and particle picking were performed in real-time using WARP v1.09[51], resulting in 4,669,932 particles, which were uploaded to cryoSPARC v3.2[52] (Supplementary Fig. 2).

Multiple rounds of 3D classification within cryoSPARC yielded a final set of 144,989 particles that gave a 3D reconstruction at a global resolution of 2.84 Å using non-uniform refinement[53]. However, since bound DNA had weak density, an alternative data processing strategy was implemented. Two rounds of ab initio reconstruction followed by heterogeneous refinement were performed on the 0° and 30° tilt datasets separately within cryoSPARC. The resulting subset of undamaged particles was combined and yielded a 3.1 Å reconstruction. These particles were then imported into Relion v3.1[54], where masks covering the core of the complex (to improve DNA density) and the bottom of the complex (to improve the quality of DrmB density) were generated within ChimeraX v1.0[55] and Relion. These masks were then used for focused 3D classification within Relion (N = 6,

T = 25). Particles within classes corresponding to DrmAB:ADP:DNA and DrmAB:ADP were then selected for Relion 3D auto-refinement and post-processing, resulting in structures with global resolutions of 3.4 Å and 3.3 Å, respectively (Supplementary Fig. 2). Additionally, A dataset of DrmAB alone was collected and processed as described above, yielding a final reconstruction with a resolution of 3.84 Å (Supplementary Fig 6).

To assist model building, the structures of multiple overlapping regions of DrmA and DrmB were predicted using the trRosetta server[56]. These models were initially fitted into the map as rigid bodies using ChimeraX. Once suitable fits were found, bulk flexible fitting was performed using Namdinator v20191016-5814c947[57]. Regions of the model that trRosetta failed to predict were either built de novo (in well-resolved regions of the map with local resolutions of up to 3.1 Å), or omitted (in flexible, poorly-resolved regions). Between the two structures, 98% of the total DrmAB sequence was modeled.

Real-space model improvement was performed using Isolde v1.2,[58] and final models were subjected to real-space refinement in Phenix v1.18rc5[59]. Structural figures were prepared using ChimeraX. Structural analysis of sequence conservation was performed using the ConSurf server[60], and visualized in ChimeraX. Motion vectors to visualize the conformational change from DrmAB:ADP to DrmAB:ADP:DNA were generated and visualized using PyMol v2.5. Schematic figure was created with BioRender.com.

**Native electrophoretic mobility shift assays (EMSAs)**. To determine the DNA substrate preferences of DrmAB, 5'Cy5-labeled 75-nt DNA was incubated with 1.5-fold excess complement, 5'block or 3'block oligos (Table 2) and heat annealed at 85 °C for 10 min prior to cooling to 4 °C. 5 nM DNA was incubated with 1 µM DrmAB or DrmA(Δloop)B in cryoEM sample buffer for 30 min at room temperature in a total volume of 20 µl. A total of 10 µl of each sample was mixed with 2 µL 6× loading buffer (30% v/v glycerol, 5% Ficoll 400, 50 mM NaCl, 10 mM HEPES pH 8, 5 mM EDTA, 0.002% w/v bromophenol blue). Electrophoresis was carried out on a non-denaturing 1.5% agarose gel for 90 min at 100 V in 1× Tris-borate-EDTA buffer. Gels were imaged using a fluorescence scanner (Fujifilm FLA-5100) with 532 nm excitation.

For comparison of DNA binding between DrmAB and DrmA(Δloop)B, serial 2-fold dilutions from of either protein complex were incubated with 5 nM Cy5-labeled DNA. Samples were incubated and EMSAs were performed as described above.

For the overhang binding assay, a 3'-Cy5-labeled DNA substrate was hybridized to unlabeled DNAs of various lengths and binding was tested as described above.

**Reporting summary**. Further information on research design is available in the Nature Research Reporting Summary linked to this article.

## Data availability

The cryo-EM structure and associated atomic model of DrmAB-ADP have been deposited into the Electron Microscopy Data Bank and the Protein Data Bank with accession codes EMD-24938 and PDB 7S9V, respectively. The cryo-EM structure of DrmAB-ADP-DNA and associated atomic model have been deposited with accession codes EMD-24939 and PDB 7S9W, respectively. Source Data are provided with this paper. Materials and correspondence requests should be addressed to Stan J.J. Brouns (stanbrouns@gmail.com) and D.W.T. (dtaylor@utexas.edu).

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

## Acknowledgements

We thank members of the Taylor and Brouns groups for comments and helpful discussions; J. Yelland in the Taylor group for assistance with the ATPase assay; and A.C. Haagsma in the Brouns group for help in protein purification. The model cartoon was created with Biorender.com. This work was supported in part by the Netherlands Organization for Scientific Research (NWO VICI; VI.C.182.027) and the European Research Council (ERC) CoG under the European Union's Horizon 2020 research and innovation program (grant agreement No. [101003229]) to S.J.J.B., and National Institute of General Medical Sciences (NIGMS) of the National Institutes of Health (NIH) R35GM138348 to D.W.T. D.W.T. is a CPRIT Scholar supported by the Cancer Prevention and Research Institute of Texas (RR160088).

## Author contributions

C.A.-M. cloned, expressed and purified all protein samples, performed and analyzed in vivo EOP assays. J.P.K.B. performed cryo-EM, structure determination, and modeling, and performed in vitro assays. J.P.K.B., C.A.-M., F.L.N., S.J.J.B., and D.W.T. analyzed and interpreted the data and wrote the manuscript. S.J.J.B. and D.W.T. supervised and secured funding for the studies.

## Competing interests

The authors declare no competing interests.
