## [Peer Review File · Nature Communications]

Structural basis for broad anti-phage immunity by DISARMEditorial Note: This manuscript has been previously reviewed at another journal that is not operating a transparent peer review scheme. This document only contains reviewer comments and rebuttal letters for versions considered at *Nature Communications*.

REVIEWER COMMENTS

Reviewer #1 (Remarks to the Author):

The authors significantly improved their manuscript by providing additional data points and articulating their mechanistic model more clearly. This reviewer (#1) is now convinced that DISARM is activated by unmethylated DNA in sequence-unspecific fashion, and that it is inhibited by methylated DNA. This lays a solid foundation for further mechanistic investigations. There are obvious unanswered questions about how the discrimination at the sensing level by DrmA/B influences the effector activity, leading to the observed functional outcome. However, this reviewer considers these remaining questions as subjects for future studies. This reviewer recommends the publication of this work without further delay.

Reviewer #2 (Remarks to the Author):

The MS by Bravo with co-authors reports two cryoEM structures of subcomplexes of DISARM, an anti-phage agent. The authors claim that they were able to identify a sensor that triggers the ssDNA binding and acts according to its status of methylation. The authors assume that this trigger loop is able to recognise a type of DNA substrates allowing binding only ssDNA or 5'ovh containing dsDNA. The results seem to be rather interesting and the structures obtained may highlight some additional info which could be useful for the following studies but certain points should be still addressed by the authors.

The authors have written that “DrmA has multiple interactions with the backbone of the DNA, mostly electrostatic contacts between positively charged side chains and DNA phosphate groups”. The authors write that they think that “non-specific electrostatic DrmAB:DNA contacts provide the molecular basis for the ... anti-phage targeting by DISARM (Ofir et al., 2017)”. However, “... conventional RM systems require recognition of specific DNA sequence motifs for nuclease activity”. It seems the last statement is essential. The authors write further that “...such specificity can be easily evaded through phage evolving escape mutations”. This idea is not consistent with the fact that the nature is very efficient and would not allow for DrmAB in the host cell (bacterium) to bind ANY ssDNA. Some specificity should exist to make self-defense of the bacterium efficient and not to harm itself, since bacteria have ssDNA plasmids in cytosol and bacteria do exchange DNA during conjugation, which is typically single stranded. So specificity for the recognition of the phage DNA is paramount for bacteria.

The authors write: "...Binding of any ssDNA triggers a significant conformational change... Domains in motion are important for the anti-phage function. Binding of an unmethylated ssDNA stimulates the ATPase activity from DmrAB, however, this stimulation is reduced if the ssDNA contains three separate methylated cytosines (i.e. the modification provided by DrmMII). These structural and biochemical observations are significant and highly novel." It is interesting, and, possibly, significant, but there are some questions.

The authors claim, that they "built near-complete atomic models of both complexes (>98% of the 220 kDa protein components), including seven nucleotides of DNA in the DrmAB:ADP:DNA complex". However they did not identify location of the "55-residue loop in DrmA". I am afraid that the authors were not able to trace the trigger loop in the DrmAB complex without DNA bound. It is rather confusing when the authors write that this loop is "absent in the DNA-bound structure"-> How can it be, if the authors have traced 98% of chains in both complex (DrmAB:ADP:DNA and DrmAB:ADP)? This loop, and differences in its position within two structures have to be identified.

There is some inconsistency within the MS. The authors did not explain if this loop represents additional non-identified so far component of the complex, or it was a part of DrmA? The authors claim that it was a loop containing 55 residues. Writing that the authors were able to trace the polypeptide chain "de novo" means that they have identified all residues of the polypeptide chains and therefore should know which ones compose this loop and if it was an additional component of the complex. This information was not provided. It was not shown even in the diagram of the sequence in figure 1. Where is this loop located? According to the schematic structural diagram (helical/loops complex) it was supposed to be in DrmA, however the figures are a bit confusing due to inconsistency in rotations of the complex in different figures: figure 3b is rather confusing: where are DrmA and DrmB : what is shown in orange, light yellow and what is in white? (a and b in Figure 3, the same problems with figure 2 a). Location of the trigger loop should be shown in figure 1 C. It seems that is located in the close proximity to DrmB.

Motion vectors showing conformational changes of DrmAB upon DNA binding. How big were these shifts? Why the distances were not measured? It would be good to see the overlay of the structures with ssDNA and without it. The extent of RMSD of the FL shift will indicate the significance of the conformational changes (if they will be bigger as a resolution of structures). The authors should be able to asses RSMD between two conformations within this area (if the structures were aligned according to the certain points like position of the DrmA. The assessable data were not provided.

Provided EM information is not reliable: to have pixel size 1.1 Angstrom at magnification of 22,550 cannot be correct. Possibly, the magnification was at least ~81,000 (read website related to the K3 cameras). Is it extremely strange why the authors were using such a high dose as ~80e/angstrom². The authors did not report which type of mode was used during the data collection: was it the super resolution mode or the integrative one? If the authors have used default settings, possibly it was the super resolution mode, but the details in processing of images were omitted: did they coarse the data? The tracing of the peptide chain was not done as "de novo": the authors have done flexible refining of the fitted in to EM maps predicted atomic models. Nothing bad is in usage such approach, but the authors have to describe it properly: the resolution and the quality of the map does not allow to do the de novo tracing chain, since the residues are not identifiable unambiguously in the shown maps.

Reviewer #3 (Remarks to the Author):

DISARM is a newly identified defense system consisting of genes encoding methylases and nuclease-related proteins, resembling the organization of typical methylation-based R-M systems. In this manuscript, the authors obtained the cryo-EM structures of the core protein DrmAB and DrmAB/DNA. The authors proposed a model in which DrmAB complex is expressed in an autoinhibited form and can be rapidly activated by specific DNA structure, i.e., 5' overhang, rather than DNA methylation to exert defensive function. DNA methylation is reported to repress the key ATPase activity, which consequently prevents autoimmune targeting. Although the detailed mechanism about how the invasive DNA is destroyed is not addressed yet, this study advances our understanding and will be of interest to the fields of phage-defence and protein-nucleic acid interactions. Since the manuscript has went through one round of revision and the authors have addressed several key questions, we have several points that I hope the authors can address them.

1. In the previous DISARM paper (10.1038/s41564-017-0051-0), Ofir et al observed that the DISARM system still protected against the modified phages in spite of their high level of methylation after propagation in the DrmMII-expressing cells. Is this observation in contradiction with the author's model? Why the methylation in phage DNA does not suppress the ATPase of DrmAB and thus impair the defense function of DISARM? The authors might discussion this in the Discussion.
2. In the class II DISARM system (10.1038/s41564-017-0051-0), the authors failed to obtain a single-gene deletion for *drmMII*, likely due to the autoimmunity. In contrast, the absence of methylase genes *drmMI* and *drmMII* does not result in autoimmunity in the class I Disarm system (10.1101/2021.12.28.474362). I have either missed something (my apologies), or the two classes have different self/nonself discrimination mechanisms although they both harbor DrmAB? If it is Yes, it would be helpful that authors can mention this difference and the possible diversity of DISARM defense mechanism?
3. The secondary structure diagram of DNA used for structural analysis displayed in Supplementary Figure 1 can be moved to figure 1.
4. It would be helpful to schematically shown the DNA substrates used in SI Figure 1e. It is hard to deduce the secondary DNA structures from Table 2.

EDITOR COMMENTS (appropriate to share with reviewers)

We thank the editor for handling this transferred manuscript. Reviewer 1 is again extremely positive and suggests publication without further delay. Reviewers 2 and 3 bring up excellent points that we believe we have thoroughly addressed in the revised manuscript. We thank the Reviewers for their comments, and we believe that they have significantly improved the clarity of the manuscript and discussion of its implications. We hope that you now find it suitable for publication in *Nature Communications*.

REVIEWER COMMENTS

Reviewer #1 (Remarks to the Author):

The authors significantly improved their manuscript by providing additional data points and articulating their mechanistic model more clearly. This reviewer (#1) is now convinced that DISARM is activated by unmethylated DNA in sequence-unspecific fashion, and that it is inhibited by methylated DNA. This lays a solid foundation for further mechanistic investigations. There are obvious unanswered questions about how the discrimination at the sensing level by DrmA/B influences the effector activity, leading to the observed functional outcome. However, this reviewer considers these remaining questions as subjects for future studies. This reviewer recommends the publication of this work without further delay.

We thank the reviewer for recognizing the novelty and significance of our work in their first review. We are very pleased they now find the manuscript suitable for publication without further delay.

Reviewer #2 (Remarks to the Author):

The MS by Bravo with co-authors reports two cryoEM structures of subcomplexes of DISARM, an anti-phage agent. The authors claim that they were able to identify a sensor that triggers the ssDNA binding and acts according to its status of methylation. The authors assume that this trigger loop is able to recognise a type of DNA substrates allowing binding only ssDNA or 5'ovh containing dsDNA. The results seem to be rather interesting and the structures obtained may highlight some additional info which could be useful for the following studies but certain points should be still addressed by the authors.

The authors have written that "DrmA has multiple interactions with the backbone of the DNA, mostly electrostatic contacts between positively charged side chains and DNA phosphate groups". The authors write that they think that "non-specific electrostatic DrmAB:DNA contacts provide the molecular basis for the ... anti-phage targeting by DISARM (Ofir et al., 2017)". However, "... conventional RM systems require recognition of specific DNA sequence motifs for nuclease activity". It seems the last statement is essential. The authors write further that "...such specificity can be easily evaded through phage evolving escape mutations". This idea is not consistent with the fact that the nature is very efficient and would not allow for DrmAB in the host cell (bacterium) to bind ANY ssDNA. Some specificity should exist to make self-defense of the bacterium efficient and not to harm itself, since bacteria have ssDNA plasmids in cytosol and bacteria do exchange DNA during conjugation, which is typically single stranded. So specificity for the recognition of the phage DNA is paramount for bacteria.

This is a good point. Given that DISARM is able to defend against a broad range of bacteriophage, logic would dictate that specificity is not incurred through recognition of a specified DNA sequence. Based on our data, we observe that unmodified DNA with a 5' single-stranded end is the preferred substrate of DrmAB. We should note that DNA binding by DISARM is weak – in our EMSAs we observe only a small fraction of the DNA (10nM) is bound by 2 μ M DrmAB complex, likely due to the presence of the autoinhibitory trigger loop.

It may be that in the absence of phage attack, DrmAB binding is out-competed by other DNA-binding proteins with much higher affinities (e.g. SSB or complexes involved in conjugative DNA transfer). It may also be the case that other factors (e.g. DrmD or DrmE) may interact with the conserved core DrmAB complex to increase DNA-binding affinity and confer specificity. Since this is speculative (and further studies are beyond the scope of the current manuscript), we have instead created a model based the data available (from previous studies and this study). This will likely set the stage for further refinements by future studies. We now add the following to our manuscript:

“Future studies are required to investigate the interplay between DrmAB and other less-conserved DISARM components (e.g. DrmD and DrmE) within DISARM, and how these subunits affect the specificity of DISARM activation.”

The authors write: “...Binding of any ssDNA triggers a significant conformational change... Domains in motion are important for the anti-phage function. Binding of an unmethylated ssDNA stimulates the ATPase activity from DmrAB, however, this stimulation is reduced if the ssDNA contains three separate methylated cytosines (i.e. the modification provided by DrmMII). These structural and biochemical observations are significant and highly novel.” It is interesting, and, possibly, significant, but there are some questions.

The authors claim, that they “built near-complete atomic models of both complexes (>98% of the 220 kDa protein components), including seven nucleotides of DNA in the DrmAB:ADP:DNA complex”. However they did not identify location of the “55-residue loop in DrmA”. I am afraid that the authors were not able to trace the trigger loop in the DrmAB complex without DNA bound. It is rather confusing when the authors write that this loop is “absent in the DNA-bound structure”-> How can it be, if the authors have traced 98% of chains in both complex (DrmAB:ADP:DNA and DrmAB:ADP)? This loop, and differences in its position within two structures have to be identified.

We apologize for the confusion. The trigger loop is within DrmA and is only present in the absence of DNA. This loop is unstructured but visible in our DrmAB:ADP and DrmAB apo structures, as shown in Supplementary Figure 4. In our DNA-bound structure, this loop likely becomes disordered, since it has been displaced by DNA, and is no longer visible. This is a similar mechanism of autoinhibition to the unrelated human DEAD-box helicase DHX37, which is cited in the paper (Singh et al., 2021).

Because we can model this loop in our 3.3 Å-resolution DrmAB:ADP map, we know exactly which residues constitute the loop (DrmA 176-232). Due to flexibility, however, we cannot unambiguously assign the register of the entire loop. We therefore chose to err on the side of caution, and model a ~25-residue region of this loop as poly-Ala. This is common practice in the field.

We have updated the schematic in Fig 1A to reflect the position of the trigger loop within the sequence of DrmA, and added the following to the figure legend:

“Presence of DrmA trigger loop (residues 176-232) is mutually exclusive with DNA-binding and is denoted by a dashed box.”

We now explicitly state the residues that constitute the TL in the relevant results section entitled “DrmA contains an unstructured trigger loop that partially occludes the DNA-binding site”.

Between the structures for DrmAB:ADP and DrmAB:ADP:DNA, we have built models that account for 98% of the total residues in DrmA and DrmB, hence we refer to our work as “near-complete atomic models”. This language is used to inform the reader that there are no large domains absent from our models. We have altered the text:

“Between these two structures, we determined atomic models that account for >98% of the total residues of the ~220 kDa protein components, with an additional seven nucleotides of DNA in the DrmAB:ADP:DNA complex”

There is some inconsistency within the MS. The authors did not explain if this loop represents additional non-identified so far component of the complex, or it was a part of DrmA? The authors claim that it was a loop containing 55 residues. Writing that the authors were able to trace the polypeptide chain “de novo” means that they have identified all residues of the polypeptide chains and therefore should know which ones compose this loop and if it was an additional component of the complex. This information was not provided. It was not shown even in the diagram of the sequence in figure 1. Where is this loop located? According to the schematic structural diagram (helical/loops complex) it was supposed to be in DrmA, however the figures are a bit confusing due to inconsistency in rotations of the complex in different figures: figure 3b is rather confusing: where are DrmA and DrmB : what is shown in orange, light yellow and what is in white? (a and b in Figure 3, the same problems with figure 2 a). Location of the trigger loop should be shown in figure 1 C. It seems that is located in the close proximity to DrmB.

As mentioned above, we have added the location of the trigger loop to the schematic in figure 1A. It is not possible to annotate its position in Figure 1C, since this is the DNA-bound structure (DNA is labelled, and the figure legend states that this is the DrmAB:ADP:ADP complex), and the loop is mutually exclusive with the presence of DNA (as is shown in Figure 3).

For the sake of clarity, we keep the same “front” view of DrmAB consistent throughout the manuscript (Figures 1c&d (left-most panel), 2a, 3a&b, and 4a). We have added a schematic to Figure 3c to indicate the direction from which the complex is being viewed, along with the following text:

“Graphic at the top right shows how the view in **c** is related to the structures in panels **a & b**.”

To further aid comprehension, we have also added the following to the figure legend:

“DrmB and parts of DrmA have been omitted for clarity.”

Figure 3b is colored consistently with all other figures in the manuscript. DrmA is burnt orange, with the RecA1 and RecA2 domains red and yellow, respectively. DrmB is white, and DUF1998 (now the HAR domain) is dark cyan. This color scheme is the one used in the schematic in Figure 1a. We now include the following in the legend for Figure 3:

“DrmAB is colored by structural domains as shown in Fig 1A”.

Motion vectors showing conformational changes of DrmAB upon DNA binding. How big were these shifts? Why the distances were not measured? It would be good to see the overlay of the structures with ssDNA and without it. The extent of RMSD of the FL shift will indicate the significance of the conformational changes (if they will be bigger as a resolution of structures). The authors should be able to assess RMSD between two conformations within this area (if the structures were aligned according to the certain points like position of the DrmA. The assessable data were not provided.

These distances were measured, and we state the following in the manuscript: “DrmB NTH **shifts ~10 Å** towards the RecA2 domain of DrmA, tightly clamping the complex around DNA.” We have added labels to the figure to demonstrate the magnitude of conformational changes. All movements <3 Å are omitted. The following has been added to the figure legend:

“Distances of conformational changes are shown parenthetically. Motion vectors for shifts smaller than 3Å are omitted.”

The RMSD between the DrmAB:ADP:DNA and DrmAB:ADP structures are very misleading since many regions of the complex do not move, thus “averaging out” the key conformational changes: DrmA aligns with 1.966Å and DrmB aligns with 1.944Å. However, from Figure 4, it is clear several regions shift by significantly more than 2Å.

Superposition of the two complexes is a good idea, We have now provided an additional Supplementary Figure to highlight the conformational changes that occur upon DNA binding:

“Supplementary Fig. 5 | Conformational changes upon DNA binding

Top: Domain architecture schematic of DrmA and DrmB. Bottom: DrmAB:ADP:DNA colored as in the above schematic, and the DrmAB:ADP complex colored in grey, showing conformational changes. DrmA trigger loop is absent in colored DrmAB:ADP:DNA model since it likely becomes disordered or flexible upon DNA binding.”

Provided EM information is not reliable: to have pixel size 1.1 Angstrom at magnification of 22,550 cannot be correct. Possibly, the magnification was at least ~81,000 (read website related to the K3 cameras).

The magnification reported is nominal and is the given magnification from the microscope. The true calibrated magnification is 45.4kx, as estimated based on ratio between the calibrated physical pixel size used for data collection (1.1Å), and the physical pixel size (5µm)

This has been updated in Supplementary Table 1.

Is it extremely strange why the authors were using such a high dose as ~80e/angstrom².

80e/Å² as a total dose is not “extremely strange”. Grant & Grigorieff 2015 use a total dose of 100e/Å² (<https://elifesciences.org/articles/6980>) to maximize the signal:noise ratio to determine high-resolution reconstructions. Our total dose is also only slightly higher than the total doses reported in “Practical considerations for using K3 cameras in CDS mode for high-resolution and high-throughput single particle cryo-EM”, which used total doses of 69e/Å².

The authors did not report which type of mode was used during the data collection: was it the super resolution mode or the integrative one? If the authors have used default settings, possibly it was the super resolution mode, but the details in processing of images were omitted: did they coarse the data?

We have updated the materials and methods to reflect that the K3 camera was in counting super-resolution mode during collection, rather than in integrating mode.

The tracing of the peptide chain was not done as “de novo”: the authors have done flexible refining of the fitted in to EM maps predicted atomic models. Nothing bad is in usage such

approach, but the authors have to describe it properly: the resolution and the quality of the map does not allow to do the de novo tracing chain, since the residues are not identifiable unambiguously in the shown maps.

We have also updated the manuscript to reflect our strategy for model building. While large amounts of DrmAB were built through flexible fitting of computationally predicted models, the resolution of the maps (which were up to 3.1Å local resolution, as is shown in Supplementary Figure 2) did indeed enable de novo building of smaller regions of the model where the predications fell short. The following has been added to the methods:

“Regions of the model that trRosetta failed to predict were either built de novo (in well-resolved regions of the map) or omitted (in flexible regions). Between the two structures, 98% of the total DrmAB sequence was modelled.”

Reviewer #3 (Remarks to the Author):

DISARM is a newly identified defense system consisting of genes encoding methylases and nuclease-related proteins, resembling the organization of typical methylation-based R-M systems. In this manuscript, the authors obtained the cryo-EM structures of the core protein DrmAB and DrmAB/DNA. The authors proposed a model in which DrmAB complex is expressed in an autoinhibited form and can be rapidly activated by specific DNA structure, i.e., 5'overhang, rather than DNA methylation to exert defensive function. DNA methylation is reported to repress the key ATPase activity, which consequently prevents autoimmune targeting. Although the detailed mechanism about how the invasive DNA is destroyed is not addressed yet, this study advances our understanding and will be of interest to the fields of phage-defence and protein-nucleic acid interactions. Since the manuscript has went through one round of revision and the authors have addressed several key questions, we have several points that I hope the authors can address them.

1. In the previous DISARM paper (10.1038/s41564-017-0051-0), Ofir et al observed that the DISARM system still protected against the modified phages in spite of their high level of methylation after propagation in the DrmMII-expressing cells. Is this observation in contradiction with the author's model? Why the methylation in phage DNA does not suppress the ATPase of DrmAB and thus impair the defense function of DISARM? The authors might discussion this in the Discussion.

This is an excellent point. While DNA methylation reduces the essential DrmAB ATPase activity required for antiphage defense, it does not completely abrogate ATP hydrolysis. We therefore propose that DISARM can still target replicating modified phage, albeit less efficiently than unmodified. We have added the following section to the discussion:

“While this model is consistent with our data, the class 2 DISARM system has been demonstrated to confer protection against modified phage. While DISARM preferentially targets unmodified DNA, methylated DNA can still support ATP hydrolysis and thus defense activation. It may be the case that DISARM activation is significantly reduced by phage DNA methylation, but the abundance of phage 5'ovh during successive cycles of replication may provide sufficient stimulus to activate DISARM. Additionally, other DISARM subunits may confer additional mechanisms to detect invading phage.”

2. In the class II DISARM system (10.1038/s41564-017-0051-0), the authors failed to obtain a single-gene deletion for drmMII, likely due to the autoimmunity. In contrast, the absence of methylase genes drmMI and drmMII does not result in autoimmunity in the class I Disarm system (10.1101/2021.12.28.474362). I have either missed something (my apologies), or the two classes have different self/nonsel self discrimination mechanisms although they both harbor DrmAB? If it is Yes, it would be helpful that authors can mention this difference and the possible diversity of DISARM defense mechanism?

This is an interesting point. Based on our current data, we cannot speculate on this point. It could be that the essential DrmE subunit in class II DISARM systems relies on drmMII DNA methylation to confer protection, whereas in class I systems (where DrmE is replaced by DrmD) such as the one used in our study does not. Since DrmAB is an essential conserved core complex between both systems, we think our model is sufficiently broad as to apply to both systems, but the importance of other subunits of the system may emerge in future studies. We have added the following to the discussion:

“Future studies are required to investigate the interplay between DrmAB and other less-conserved DISARM components (e.g. DrmD and DrmE) within DISARM, and how these subunits affect the specificity of DISARM activation.”

3. The secondary structure diagram of DNA used for structural analysis displayed in Supplementary Figure 1 can be moved to figure 1.

We have done this.

4. It would be helpful to schematically shown the DNA substrates used in SI Figure 1e. It is hard to deduce the secondary DNA structures from Table 2.

We have added this to Supplementary Figure 1. The only structured DNA substrates are the one used for structural studies “DNA stem loop” and the methylated DNA “Methylated DNA stem loop”. These are now depicted in Figure 1c. We have additionally added schematics to illustrate the overhang experiments in Supplementary Figure 1e.

REVIEWERS' COMMENTS

Reviewer #2 (Remarks to the Author):

The MS has been corrected a bit, main comments given in the rebuttal letter, while being useful and helpful, are not incorporated into the MS.

In the rebuttal letter the authors have written that magnification was about 45,000 but in the table it is still 22,500, which is wrong.

The authors have written in the rebuttal letter that they have done flexible fitting of the model predicted in Rosetta, however in the methods it was still "de novo" fitting, which is wrong. The answers given in the rebuttal letter has to be incorporated to the MS.

The authors should not mislead a reader. Unfortunately, the structure while being interesting does not have a resolution of 3.2 Angstrom, it is about 3.8 Angstrom, which is OK, but it prevents doing de novo fitting.

Reviewer #3 (Remarks to the Author):

The authors had addressed all the concerns raised about the last submission. This is a nice and important work. I support publication.

Response to Reviewers

Reviewer #2 (Remarks to the Author):

The MS has been corrected a bit, main comments given in the rebuttal letter, while being useful and helpful, are not incorporated into the MS. In the rebuttal letter the authors have written that magnification was about 45,000 but in the table it is still 22,500, which is wrong.

We thank the reviewer for pointing out the discrepancy in magnifications reported. We have updated Supplementary Table 1.

We have incorporated the necessary changes in the manuscript. The first rebuttal letter was just clarifying information that was already present or was modified during the first review: in the main text of the manuscript, Methods, or Supplementary Information.

The authors have written in the rebuttal letter that they have done flexible fitting of the model predicted in Rosetta, however in the methods it was still "de novo" fitting, which is wrong. The answers given in the rebuttal letter has to be incorporated to the MS. The authors should not mislead a reader. Unfortunately, the structure while being interesting does not have a resolution of 3.2 Angstrom, it is about 3.8 Angstrom, which is OK, but it prevents doing de novo fitting.

We do not mislead the reader in any way. The information required for the reader to analyze the data is in the manuscript and associated files.

In Supplementary Figure 2, we include gold-standard Fourier Shell Correlation (FSC) plots for the DrmAB:ADP and DrmAB:ADP:DNA reconstructions that clearly show resolutions of 3.3 Å and 3.4 Å, respectively. This figure has been an integral aspect of all iterations of our manuscript. We have never claimed that our structure has a global resolution of 3.2 Å.

The global resolution of the DrmAB:ADP:DNA *model* is 3.8Å, as determined by the resolution at which the map-to-model FSC curve drops below the 0.5 threshold. This metric is not the resolution of the map itself, since this metric exclusively reflects the overall quality of how well the map reflects the model. The map-to-model metric does not reflect how interpretable the map is – it reflects how well the entire model corresponds to the entire map. It is incorrect to assert that the map is of unsuitable quality to build small, well-defined regions *de novo*.

Complexes with large, flexible regions will have a reduced map-to-model FSC, regardless of how well-resolved other regions are. This is indeed the case with our complex. Therefore, we decided to use trRosetta to predict the structures of fragments of our model, which we could dock and then flexibly fit within our map (improved tools such as AlphaFold2 were not available at the time). For regions of our model that trRosetta failed to accurately predict, we instead had to use an alternative strategy, which depended on the local resolution of the corresponding region of the map.

If the local resolution was of high quality (up to 3.1 Å in many regions at the core of the complex – please refer to Supplementary Figure 2d) then we could confidently build *de novo*. This is reflected in our new supplementary figure (new Supplementary Figure 3), which shows representative densities of alpha helices, beta-sheets, ADP and DNA. Within many regions, densities corresponding to side-chains are excellently resolved, allowing modelling *de novo* of regions of the complex. However, if the local resolution of a given region was poorly resolved and trRosetta failed to predict a suitable model, then that region of the model was omitted. The omitted regions account for a total of ~2% of the entire DrmAB complex model. Our approach is not only standard in the field but is an example of good practice – our models are reflective of what we could confidently interpret from our maps. To characterize this approach as otherwise is simply incorrect.

We claim that small, well-resolved regions of the model were “built *de novo*,” and in our cryo-EM modeling statics Supplementary Table 1 (initial PDB file used: *de novo*). In this context, it means that there were no models in the Protein Data Bank (PDB) that we could use for modeling. We have changed this section of the table to “N/A” for the sake of clarity.

Additionally, for the sake of clarity, we have amended the Methods section accordingly:

“Regions of the model that trRosetta failed to predict were either built *de novo* (in well-resolved regions of the map with local resolutions of up to 3.1Å), or omitted (in flexible, poorly-resolved regions).”

As mentioned above, we have now added a new, additional figure to the supplementary materials (Supplementary Figure 3) showing representative cryo-EM densities for our DrmAB:ADP:DNA map.

Reviewer #3 (Remarks to the Author):

The authors had addressed all the concerns raised about the last submission. This is a nice and important work. I support publication.

We thank the reviewer for their feedback. We feel that their previous comments have improved the quality of our manuscript.